# Simultaneous Care in Oncology: A 7-Year Experience at ESMO Designated Centre at Veneto Institute of Oncology, Italy

**DOI:** 10.3390/cancers14102568

**Published:** 2022-05-23

**Authors:** Antonella Brunello, Antonella Galiano, Stefania Schiavon, Mariateresa Nardi, Alessandra Feltrin, Ardi Pambuku, Chiara De Toni, Alice Dal Col, Evelina Lamberti, Chiara Pittarello, Francesca Bergamo, Umberto Basso, Marco Maruzzo, Silvia Finotto, Maital Bolshinsky, Silvia Stragliotto, Letizia Procaccio, Mario Domenico Rizzato, Fabio Formaglio, Giuseppe Lombardi, Sara Lonardi, Vittorina Zagonel

**Affiliations:** 1Department of Oncology, Medical Oncology 1, Veneto Institute of Oncology IOV-IRCCS, Via Gattamelata 64, 35128 Padua, Italy; antonella.brunello@iov.veneto.it (A.B.); antonella.galiano@iov.veneto.it (A.G.); chiara.detoni@iov.veneto.it (C.D.T.); evelina.lamberti@iov.veneto.it (E.L.); chiara.pittarello@iov.veneto.it (C.P.); francesca.bergamo@iov.veneto.it (F.B.); umberto.basso@iov.veneto.it (U.B.); marco.maruzzo@iov.veneto.it (M.M.); silvia.finotto@iov.veneto.it (S.F.); maital.bolshinsky@iov.veneto.it (M.B.); letizia.procaccio@iov.veneto.it (L.P.); mariodomenico.rizzato@iov.veneto.it (M.D.R.); giuseppe.lombardi@iov.veneto.it (G.L.); 2Pain Therapy and Palliative Care Unit, Veneto Institute of Oncology IOV-IRCCS, Via Gattamelata 64, 35128 Padua, Italy; stefania.schiavon@iov.veneto.it (S.S.); ardi.pambuku@iov.veneto.it (A.P.); alice.dalcol@iov.veneto.it (A.D.C.); fabio.formaglio@iov.veneto.it (F.F.); 3Clinical Nutrition Unit, Veneto Institute of Oncology IOV-IRCCS, 35128 Padua, Italy; mariateresa.nardi@iov.veneto.it; 4Hospital Psychology, Veneto Institute of Oncology IOV-IRCCS, 35128 Padua, Italy; alessandra.feltrin@iov.veneto.it; 5Department of Oncology, Medical Oncology 3, Veneto Institute of Oncology IOV-IRCCS, 31033 Castelfranco Veneto, Italy; silvia.stragliotto@iov.veneto.it (S.S.); sara.lonardi@iov.veneto.it (S.L.)

**Keywords:** simultaneous care, early palliative care, indicators of integration, advanced disease, symptom control

## Abstract

**Simple Summary:**

At the Veneto Institute of Oncology-IRCCS, a simultaneous-care outpatient clinic (SCOC) has been active since 2014. Here, patients with advanced-stage disease are evaluated by a multidisciplinary team composed of an oncologist, a palliative care physician, a physician specialized in clinical nutrition, a psycho-oncologist and a nurse navigator, offering an early approach to palliative care, as suggested by clinical and scientific evidence. At the end of 2017, a procedure was implemented with the definition of indicators collected annually to evaluate the performance of the SCOC. This study aimed to describe the activity of the SCOC over the years, as well as its performance through the evaluation of these indicators. This is the first report which analyzed the effectiveness of an outpatient clinic where the patient is evaluated simultaneously by the oncologist and the palliative care team and illustrates a new organizational model to improve good clinical practice.

**Abstract:**

Benefits of early palliative care referral in oncology are well-validated. At the Veneto Institute of Oncology-IRCCS, a simultaneous-care outpatient clinic (SCOC) has been active since 2014, where patients with advanced cancer are evaluated by an oncologist together with a palliative care team. We prospectively assessed SCOC patients’ characteristics and SCOC outcomes through internal procedure indicators. Data were retrieved from the SCOC prospectively maintained database. There were 753 eligible patients. The median age was 68 years; primary tumor sites were gastrointestinal (75.2%), genitourinary (15.0%) and other sites (9.8%). Predominant symptoms were psychological issues (69.4%), appetite loss (67.5%) and pain (65.9%). Dyspnea was reported in 53 patients (7%) in the referral form, while it was detected in 226 patients (34.2%) during SCOC visits (*p* < 0.0001). Median survival of patients after the SCOC visit was 7.3 months. Survival estimates by the referring oncologist were significantly different from the actual survival. Psychological intervention was deemed necessary and undertaken in 34.6% of patients, and nutritional support was undertaken in 37.9% of patients. Activation of palliative care services was prompted for 77.7% of patients. Out of 357 patients whose place of death is known, 69.2% died at home, in hospice or residential care. With regard to indicators’ assessment, the threshold was reached for 9 out of 11 parameters (81.8%) requested by the procedure. This study confirmed the importance of close collaboration between oncologists and palliative care teams in responding properly to cancer patients’ needs. The introduction of a procedure with indicators allowed punctual assessment of a team’s performance.

## 1. Introduction

Mounting evidence from randomized trials and meta-analyses supports early integration of palliative care for patients with advanced-stage cancer in the treatment trajectory [1]. In particular, early palliative care has been shown to improve symptom control, quality of life, patient and caregiver satisfaction, quality of end-of-life care, costs of care and in some cases also survival in cancer patients [2,3,4,5,6]. Therefore, early palliative care is now acknowledged by most prominent international oncology scientific societies and recommended in their guidelines (European Society of Medical Oncology—ESMO [7]; American Society of Clinical Oncology—ASCO [8], National Comprehensive Cancer Network—NCCN [9]) as well as by the Italian Association of Medical Oncology—AIOM [10].

Nevertheless, to date there is no optimal and shared model that can guarantee the best procedure. Several models of early integration have in fact been developed, which may be classified based on the setting of care and method of palliative care referral. Bruera and Hui described three models of integrated care of oncology and palliative care, emphasizing shared care in which the oncologist routinely refers patients to a specialist palliative care team early on in the disease trajectory and collaborates closely with the interdisciplinary palliative care team [11]. In 2015, an international consensus identified the 13 leading indicators of oncology and palliative care integration [12]. These indicators need to be fulfilled by the ESMO Designated Centers (DC) of Integrated Oncology and Palliative care program [13].

One of the main objectives of early palliative care is indeed sharing the care path for the whole trajectory of an illness between the oncologist, palliative care team and primary care physician. The goal is therefore to connect and communicate between “silos” in a transversal way, to take care of the patient in a complete and shared way, as proposed by a Lancet oncology commission in 2018 [14].

Recently, Hui et al. suggested that, in order to identify patients with high supportive care needs and to refer these individuals to specialist palliative care in a timely manner, a systematic process based on standardized referral criteria is required, which involves four elements: (1) routine screening of supportive care needs at oncology clinic evaluations; (2) establishment of institution-specific consensual criteria for referral; (3) a systematic procedure that triggers referral when patients meet criteria and (4) availability of outpatient palliative care resources to deliver personalized, timely patient-centered care [15].

The Veneto Institute of Oncology (IOV) is an Italian national comprehensive cancer center which takes care of more than 5000 new cancer patients per year, and offers preventive, curative and palliative care services. Early integration of palliative care in the oncological care pathway has been a prerogative and a consolidated model for the care of patients since 2012, when the Oncology Department was recognized by ESMO as an ESMO-DC. In 2014, a simultaneous-care outpatient clinic (SCOC) was activated in which patients with advanced-stage disease are evaluated by a multidisciplinary team composed of an oncologist, a palliative care physician, a physician specialized in clinical nutrition, a psycho-oncologist and a nurse navigator. The involved oncologist was double-board-certified in medical oncology and palliative medicine. Moreover, most palliative care specialists are also certified as medical oncologists. In order to define timely palliative care referral [16] through objective criteria for access to the SCOC, as well as to establish priority of access according to NCCN criteria [17] and the international consensus [18], in January 2018, a referral form for identifying patients with palliative care needs was defined by oncology and palliative care teams (Figure A1). Oncologists fill out the form at the time of their visit. Items included in the form are: Karnofsky Performance Status, symptoms, estimated survival, availability of cancer treatments with impact on survival, expected toxicity from treatment and social problems. Items are scored, with the total score ranging from 1 to 21, with a higher score indicating a greater palliative care need. A Karnofsky score ≤ 50 was chosen as the cut-off since it is indicative of poor performance status and associated with a higher need for palliative care, and, despite having a certain subjective variability, several studies have demonstrated a better reliability of the Karnofsky index compared to ECOG in estimating patient performance status [19,20]. The priority for access of patients to the SCOC is based on the final score. Patients are informed by their oncologist about the benefit of palliative care and are strongly recommended to attend the SCOC visit. Visits at the SCOC are scheduled every 45 min. During the visit, the symptom burden is investigated, the degree of awareness of diagnosis and prognosis is assessed, and the oncologic treatment program as well as the decisions on how to proceed are shared. Patients therefore continue oncologic care, and through advance care planning, receive follow-up procedures from the other specialists based on the identified needs (Figure 1). 

In addition, at the end of 2017, a procedure was implemented with the definition of indicators collected annually to evaluate the performance of the SCOC activity. 

This study aimed to describe the activity of SCOC over the years as well as its performance through the evaluation of 11 indicators (2 appropriateness indicators, 4 process indicators and 5 outcome indicators).

## 2. Patients and Methods

### 2.1. Patients 

The study was conducted at the IOV, Padua, Italy. Patients referred to SCOC by Oncology Unit 1 between 1 January 2018 and 31 December 2021 (time frame since the referral form was adopted) were eligible for the study. 

Data were retrieved from the SCOC prospectively maintained database. The following data were collected:Demographic information (age, sex), tumor site and extension.Referral form scores. Scoring systems in the adopted referral form classify patients into three priority groups: (a) visit needs to be scheduled within 15 days in case of score ≥10 (high score); (b) patients with a score between 5 and 9 should be scheduled for a visit within 1 month (intermediate score); and (c) patients with a score between 0 and 4 visit should be scheduled for a visit within 2 months (low score).Timing of referral in relation to diagnosis and to death.Symptom burden assessed by ESAS (Edmonton Symptom Assessment Scale). ESAS is a measure of symptom burden that includes a Likert rating of nine symptoms (pain, fatigue, drowsiness, nausea, anxiety, depression, appetite loss, dyspnea, wellbeing) on a scale from 0 (best) to 10 (worst), with a cumulative score of 0–90 [21], which has been adopted for routine needs screening during SCOC visits.Need and opportunity for psychological intervention, as per ESAS results.MUST (Malnutrition Universal Screening Tool) score [22,23]. MUST identifies patients who are malnourished or are at risk of malnutrition; the score ranges from 0 to 6, with 0 = low risk of malnutrition, 1 = medium risk and ≥2 = high risk.Need and opportunity of nutritional intervention.Patient’s awareness of diagnosis and prognosis.Advance care planning and survival estimated by the oncologist at the time of referral.

Flow chart of the study is shown in Figure 2. Out of 1625 patients evaluated in SCOC since March 2014, 819 patients enrolled for the study. Of these, 66 were excluded because no cancer-directed treatment was planned, with the number of total eligible patients being 753. 

### 2.2. Statistical Analysis 

Patients’ characteristics were described by descriptive analysis. The comparisons were tested using chi-squared tests, proportions tests, Kruskal–Wallis tests and log-rank tests, as appropriate. For the survival analysis, all patients entered the study at the date of their visit to the SCOC and were followed up until 31 January 2022 or the date of death. The Kaplan–Meier method was used to calculate and draw the cumulative survival rates. R Version 4.1.2 was used to perform all statistical analyses. The level of significance was set at 5%.

## 3. Results

Patients’ characteristics are shown in Table 1. The median age was 68 years (IQR: 60–76). A total of 566 patients (75.2%) had gastrointestinal cancer (GC), 113 (15.0%) had urological cancer (UC), and 74 (9.8%) had other types of cancer (OC) (namely sarcoma, breast, lymphoma and gynecological cancer). The vast majority of patients (90.9%) had metastatic disease at the time of their visit to the SCOC; 338 patients (44.9%) had received one line of systemic anticancer therapy, 192 (25.5%) two lines and 223 (29.6%) more than two. Median time from cancer diagnosis to SCOC referral was 11.5 months (3.5–26.9), and was significantly shorter for patients with gastrointestinal malignancies (9.1 months) compared to patients affected by genitourinary malignancies (27.8 months) and other types of cancers (13.2 months) (*p* < 0.0001). For 76 patients (10.1%), data on the time interval from cancer diagnosis to SCOC visit were missing.

Based on the characteristics detected by the referral form, 144 patients (19.1%) had a high score, 533 patients (70.8%) an intermediate score, and 76 patients (10.1%) had a low score, with a median score of 7 (IQR: 6–9). Karnofsky Performance Status was ≥70 in 87.8% of patients.

The comparison of symptoms between the referral form and SCOC visit was evaluated. The predominant symptoms in the referral form were appetite loss (558, 74.1%) and weight loss (466, 61.9%). The most frequent symptoms at the time of the SCOC visit were appetite loss and psychological disorders (anxiety or depression, or both), which were present in 453 (67.5%) and 460 (69.4%) patients, respectively. Pain was present in more than half of the patients (referral form: 56.3%, SCOC visit: 65.9%). Dyspnea was reported in 53 patients (7.0%) in the referral form, while it was detected in 226 patients (34.2%) during the SCOC visit (*p* < 0.0001). Social or welfare problems were present in 30 patients (4.0%) (Figure A2).

As of 31 December 2021, 551 patients (73.2%) are deceased. Median survival of the whole group of patients from the date of the SCOC visit was 7.3 months (range: 6.5–8.0). Overall survival was significantly different in the three groups of cancer types, with median survival being 6.6 months in the GC group, 7.4 months in patients with UC and 10.1 months for OC (*p* < 0.0001). 

Figure 3 compares the survival estimated by the oncologist at the time the referral form was filled in with the actual survival. Survival ranges were categorized into three categories: less than 6 months, between 6 and 12 months and more than 12 months, which are consistent with the referral form. Differences between estimated and observed survival were statistically significant in all three prognostic groups (<0.0001, <0.0001, 0.0218, respectively).

ESAS score results are shown in Table 2 (Figure A3). Missing values for some symptoms were present in up to 92 patients. The median ESAS score was 23 (range: 0–84). The majority of patients had a low score, yet 76.4% of the patients had four or more burden symptoms. A total of 430 (63.4%) patients had at least one symptom with an intensity of 7–10, and of these, 175 (40.7%) had three or more symptoms with an intensity of 7–10. The median ESAS score in this group was 33 (range: 8–84).

### 3.1. Evaluation of Indicators

Performance indicators as per the SCOC procedure and their threshold, along with the monitoring results, are reported in Table 3.

#### 3.1.1. Appropriateness Indicators

With regard to appropriateness indicators, 91.9% patients were undergoing active oncological treatment at the time of the SCOC visit. The proportion of referred patients with survival <6 months increased over time from 40.3% in 2018 to 65.9% in 2021 (*p* < 0.0001) (Table 4). We also assessed the trend over the years of patients referred to the SCOC according to the form score (Figure 4); patients with a score ≥10 increased over the years from 30 (15.3%) in 2018 to 47 (21.3%) in 2021 (*p* = 0.0941). 

#### 3.1.2. Process Indicators

With regard to the process indicators, the total number of patients referred to the SCOC had a referral form completed by their oncologist. As for the scheduled time of the SCOC visit according to the referral form score, out of 671 evaluable patients, 456 (68.0%) had an SCOC visit within the preplanned time frame; in particular, as shown in Table 5, 96 patients (75.0%) scoring ≥10 had the SCOC visit within 15 days, 309 (64.8%) of those with scores 5–9 had the visit within 1 month, and 51 (77.3%) of those with a low score had the visit scheduled within 2 months, as planned.

Every patient received a report or advance care planning at the end of their SCOC visit.

#### 3.1.3. Outcome Indicators

Table 6 shows the output of the SCOC visit. Psychological assessment was carried out for all patients. A total of 643 (95.3%) had full awareness of their cancer diagnosis, of whom 357 (58.5%) also had a full awareness of the prognosis. Psychological problems were detected in 286 patients (38.0%), of whom 99 required care from the psychologist. Depression was the most prevalent issue (67.1%). 

As for the nutritional assessment, the MUST score was evaluated for 652 patients (86.6%). Of these, 44.2% had no nutritional problems, and 117 (17.9%) had a MUST score of 1. Two hundred forty-seven patients (37.9%) had a MUST score in the 2–6 range (high risk of malnutrition) and received nutritional support. Among these, 184 (74.5%) had GC, and 58.3% died within 6 months. A total of 113 (63.5%) showed weight stabilization after nutritional support for 30 days after the SCOC visit, while 31 subjects (17.4%) lost weight, and 34 (19.1%) had gained weight. For 69 (27.9%) patients with a MUST score ≥ 2, follow-up data for weight were not available. 

On the basis of the needs detected by the ESAS score, activation of palliative care services was undertaken for 585 patients (77.7%), in 91 patients (15.6%) from the Palliative Care Unit of the Institute and in 461 patients (78.8%) through activation of territorial services. Of the latter, for 5.6% of patients, activation of territorial services was postponed by the primary care physician. 

Patients with unplanned emergency room admissions after the SCOC visit totaled 170 (22.6%), of whom 59 patients (7.8% of total population) had more than one admission, and 111 had a single admission.

Finally, with regard to the place of death of the 357 patients for which this is known, 247 (69.2%) died in a non-hospital setting (37.8% at home, and 30.3% in hospice and 1.1% in residential care), and 110 (30.8%) died in hospital (Figure 5).

Globally, the threshold was reached for 9 out of 11 parameters (81.8%), as requested by procedure.

## 4. Discussion

Early integration of palliative care in the cancer patient’s journey is today regarded as an essential goal to optimize quality of life, especially in advanced stages of disease [15,24]. While there is international agreement that the patient should be referred early, there is no “one size fits all” model [25]. Every center, in fact, in relation to structure, organization and service availability, should find the most appropriate way to adequately respond to patients’ needs [15,26,27]. Measuring their performance in service delivery in order to quantify the level of integration of internal services is of the utmost importance for each hospital. This, in turn, would allow patients and clinicians to identify centers of excellence for palliative care; moreover, it would allow for identifying quality improvement opportunities for integration as well as measuring progress over time [13]. Integration is a long and complex process, with several areas requiring integration. The five main categories that identify the level of integration are: research, education, process, administration and infrastructure [1]. Activation of an SCOC requires a practice change within oncology departments that allows for close/full collaboration and sharing procedures for integrated care paths. 

The procedure put in place in our department meets all four criteria proposed by Hui et al. to ensure timely activation of palliative care [15]. In addition, our SCOC plan meets the development, feasibility, evaluation and implementation criteria proposed by Zimmermann for building an effective early palliative care team [28]. Furthermore, a predefined procedure which includes annual detection of indicators has allowed us to assess the performance and to improve service organization. Indeed, to the best of our knowledge, this is the first report which analyzes the effectiveness of a multidisciplinary outpatient clinic where the patient is assessed simultaneously by the oncologist and the palliative care team. This innovative modality allows for a direct interaction between specialists regarding treatment opportunities, patient survival, awareness of the patient’s state of health and presence or absence of a caregiver. It also guarantees a softer approach for the patient to palliative care, accompanied by the oncologist. Embedding a full palliative care team within the oncology clinic has the potential advantage of improving the volume and timeliness of referral and reinforcing multiprofessional growth through optimization of communication between the oncology and palliative care team, while also maximizing convenience for patients [1]. This SCOC is highly appreciated by patients and allows for unique care within a realistic time frame and communication to the patient about his overall state of health and perspectives of care.

The SCOC allows for a more accurate assessment, through the ESAS, of the frequency and intensity of symptoms in all patients. In particular, dyspnea was reported in 53 patients (7.0%) in the referral form, while it was detected in 226 patients (34.2%) during their SCOC visit (*p* < 0.0001). Indeed, while toxicities and laboratory results can be consistently reported by healthcare personnel, subjective experiences such as symptoms are best reported by patients themselves, and several studies demonstrated under-reporting of symptoms by clinicians compared to patient self-reporting [29,30,31]. Patient-reported outcome measures, such as the ESAS in the advanced disease setting, are therefore fundamental for optimal acknowledgement of symptoms, and their proper management, improving quality of life, and possibly survival. In our study, in fact, symptom screening by the ESAS allowed for the identification of patients with greater symptom burden who would benefit most from early palliative care [32].

Considering the numbers of patients admitted during the period (on average 2000 new patients/year), about 9% of patients were evaluated per year in the SCOC by the access form designed to prioritize patients with greater and more urgent needs. The use of a score form in which four criteria are needs-based and two prognosis-based has allowed for the identification of a subgroup of patients with a limited prognosis and with a major burden of symptoms, who were quickly taken into care according to their needs. In fact, both the prognosis-based and the needs-based referral identify a patient population who can greatly benefit from specialized palliative care [33]. 

We have observed an improvement in patient referral over the years (increase in patients with higher scores), and the percentage of patients with life expectancy <6 months has also increased. Visits for patients with less urgent needs are instead planned with a longer time interval, while still ensuring access to the SCOC outpatient clinic. 

Palliative care services were activated for 77.7% of the patients, yet in 5.6% of cases, this occurred some time after the request because of the primary care physician’s related issues. About 38% of patients required nutritional support, and a similar proportion of patients manifested psychological problems. About one-fourth of patients (22.6%) had an unplanned admission to the emergency room after SCOC evaluation, even several times, although this was seen for a minority of patients. 

With regard to indicators’ assessment, thresholds set in the procedure were not reached in two parameters. In detail, SCOC visits were not undertaken within the time frame for all the patients. This was due to patient request in some cases (i.e., for logistical problems); in other cases, this was due to unavailability of SCOC appointments. As for the place of death, unfortunately, this did not always occur where expected. This parameter could be influenced by delayed activation of territorial palliative care services, as well as by logistical family situations [34]. In fact, although we know that the majority of cancer patients wish to die at home [35], there are many factors that prevent this from happening [34,36]. Certainly, the hospital is not considered suitable, although it is still requested by as many as 17% of patients [35], likely due, at least in part, to lack of knowledge of hospice as a dedicated environment for definitive palliative care. In fact, surprisingly, a recent study in which patients received palliative care at home showed that 16.7% of them died in the acute care hospital [34]. Moreover, a home death requires an adequate socio-economic situation, good symptom control and availability of services, and some patients with less financial or social resources may not be candidates for home end-of-life care and death.

Our study confirmed an overestimation of prognosis by oncologists, as well as an excessive expectation for the efficacy of cancer therapies in advanced disease, even though many oncologists have substantial palliative care education. Surprisingly, these parameters were homogeneous across tumor types, underlying a univocal approach of the oncologists of the same oncology unit. Continuing education of medical oncologists in palliative care remains critical for both providing the first level of palliative care [14] and facilitating early access to the integrated SCOC [13,37]. As reported in the literature, oncologists’ referral practices can change if positive consequences of early referral are demonstrated [38]. Additionally, support from oncologists is crucial in order to encourage patients to attend their scheduled appointment [26,39].

Our study also confirmed the benefit of an early palliative care approach in patients with solid tumors, as also reported in another Italian experience by Bandieri et al., in terms of improved pain control [40] as well as a number of other parameters, including a decrease in the aggressiveness of chemotherapy at the end of life [41]. Similar findings have also been recently confirmed in the oncohematology setting by the same Italian group [42]. 

## 5. Strengths and Limitations

Several limitations to this study must be mentioned. First, data collection was limited to a single center, which restricts the extrapolation of results to the general population. Second, the choice of parameters in the assessment form was based on data from the literature and on the intersociety document shared at national level by the AIOM (Italian Medical Oncology Association) and SICP (Italian Society of Palliative Care) [43], yet the scoring systems of the various parameters were arbitrarily attributed. Third, given the observational nature of this study, it was not possible to evaluate the effectiveness of this approach compared with a control group. Fourth, factors related the oncologist’s reasons for referral were not evaluated, although they would provide additional information. Lastly, we should also point out that our study did not evaluate any differences in cost, patient preference or outcomes for the embedded model that we used, compared with the more traditional model of patient referral to a palliative care service. Embedded versus independent outpatient clinics need to be studied in prospective studies.

This study has two major implications for clinical practice. Firstly, it supports the use of a dynamic interaction between oncology and specialist palliative care team in order to provide patients with service customized to their needs. The presence of the oncologist in the SCOC can facilitate the patient’s approach to palliative care and allows for direct sharing among the palliative care team regarding treatment options, life expectancy and patient awareness of prognosis. Secondly, our results reinforce the belief that it is possible to improve good clinical practice through new organizational models.

## 6. Improvement Actions

The outcome of this study and performance evaluation has allowed us to share some decisions to improve the service, such as: (a) continuing education of oncologists in palliative care (prognosis assessment); (b) expanding availability of access to SCOC outpatient clinic; (c) training and sharing with family physicians; (d) verifying the activation of territorial palliative care services within the timeframe established by the SCOC; and (e) administering patient a questionnaire on perceived quality of the SCOC. 

## 7. Conclusions

This study confirmed the importance of close collaboration between oncologists and palliative care teams to ensure that all cancer patients receive an early approach to palliative care. The introduction of one procedure with indicators allowed us to evaluate the performance of the team as well as to intervene through continuing education and sharing, to improve the organization of the service, to implement accessible modalities and motivate oncologists to refer SCOC patients early, based on needs and life expectancy. Greater integration with home-care services (primary care physician and palliative care services) is necessary to ensure timely care of patients with high burden needs in palliative care, avoiding unplanned admissions to the emergency room and ensuring that death occurs, when feasible, in the patient’s preferred location.

## Figures and Tables

**Figure 1 cancers-14-02568-f001:**
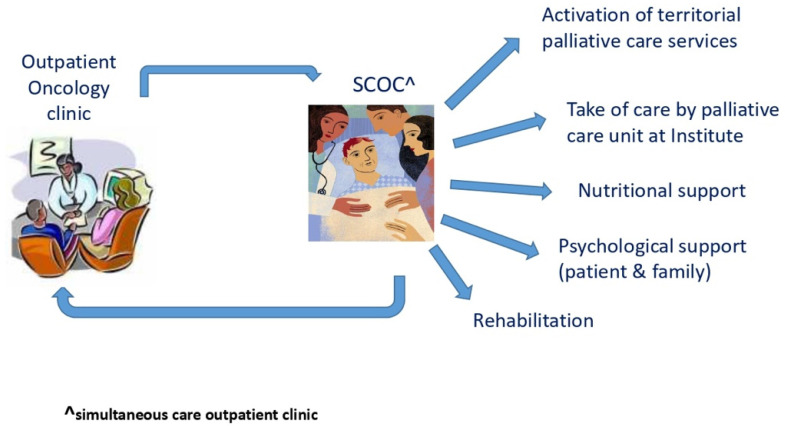
Patient journey.

**Figure 2 cancers-14-02568-f002:**
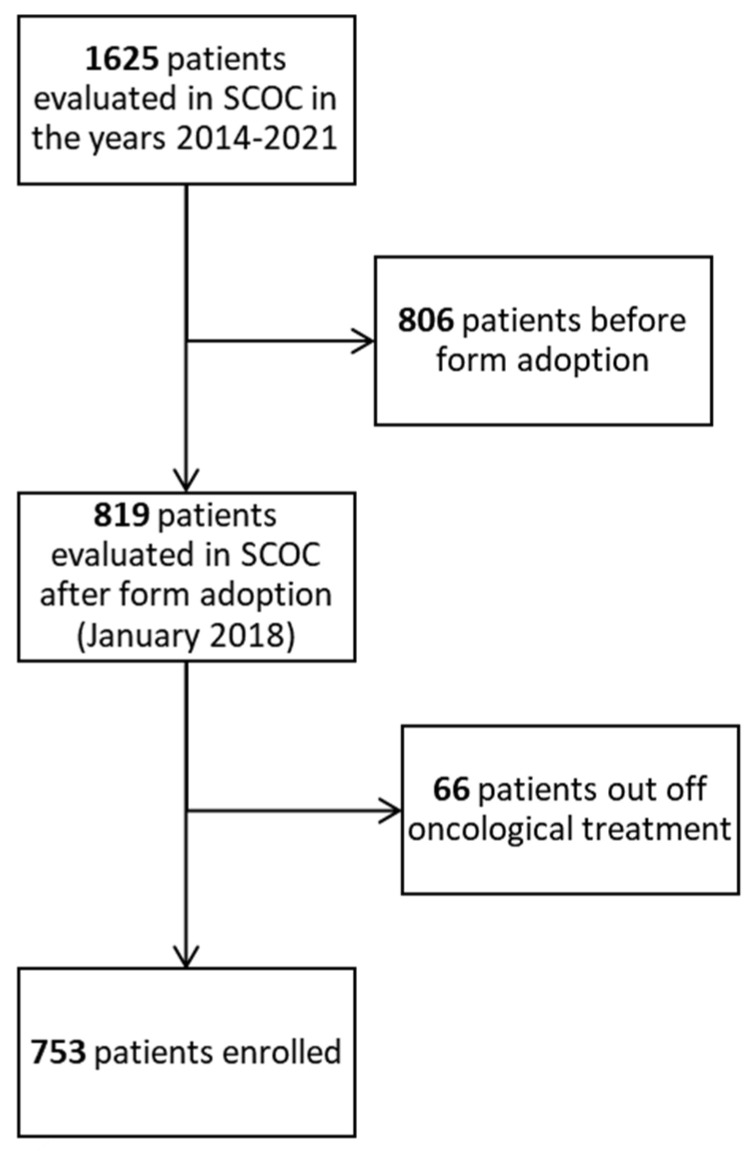
Flow chart of the study.

**Figure 3 cancers-14-02568-f003:**
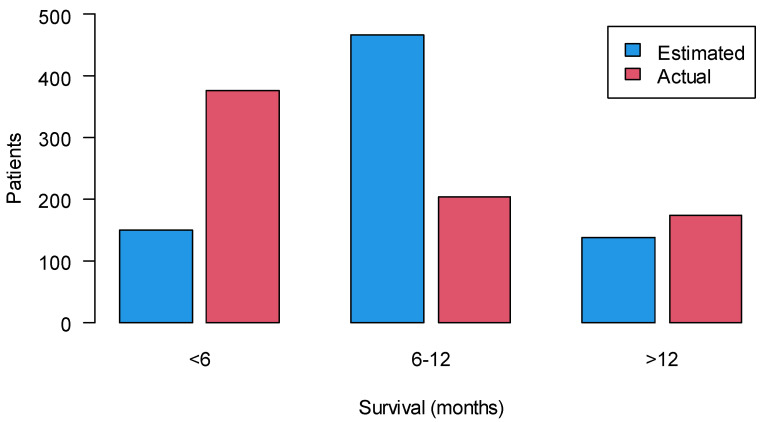
Comparison between actual and estimated survival.

**Figure 4 cancers-14-02568-f004:**
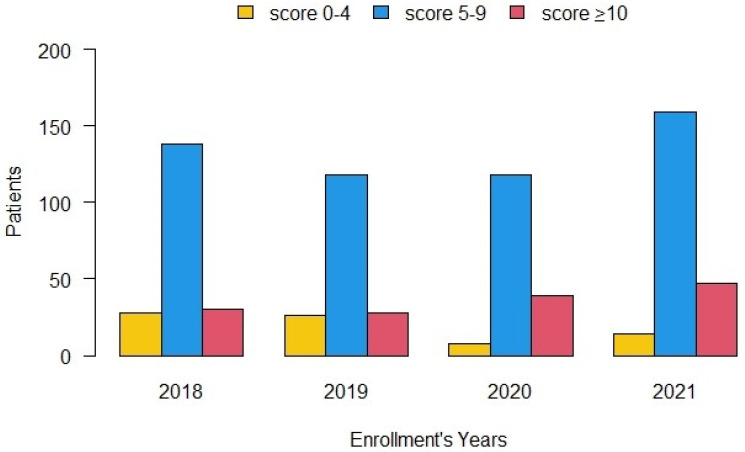
Enrollment of patients according to referral form score and year of referral.

**Figure 5 cancers-14-02568-f005:**
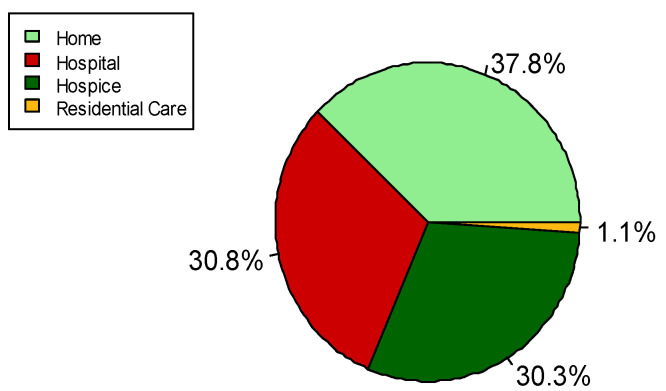
Pie chart of place of death. Data available for 357 subjects.

**Table 1 cancers-14-02568-t001:** Patients’ characteristics.

Characteristics	N Patients	%
Gender		
Male	435	57.8
Female	318	42.2
Age at referral (years)	68 (median)	60–76 (IQR)
Tumor site		
Gastrointestinal (GI)	566	75.2
-Upper GI	120	21.2 *
-Colorectal cancer	215	38.0 *
-Hepatobiliopancreatic	231	40.8 *
Urological	113	15.0
Other (sarcoma, lymphoma, gynecological)	74	9.8
Tumor stage		
Locally advanced	47	6.2
Metastatic	684	90.9
Missing	22	2.9
Treatment line		
First-line	338	44.9
Second-line	192	25.5
Third or further lines	223	29.6
Years since cancer diagnosis		
≤1	351	51.8
>1	326	48.2
Enrollment year		
2018	196	26.0
2019	172	22.9
2020	165	21.9
2021	220	29.2

* Percentages calculated on the gastrointestinal group.

**Table 2 cancers-14-02568-t002:** Frequency distribution for the ESAS score.

Symptom	Score					
	N pts	(%)	N pts	(%)	N pts	(%)
	0–3		4–6		7–10	
	N pts	(%)	N pts	(%)	N pts	(%)
**Pain**	392	(58.1)	150	(22.2)	133	(19.7)
**Fatigue**	150	(22.3)	243	(36.1)	281	(41.7)
**Nausea**	522	(78.0)	85	(12.7)	62	(9.3)
**Depression**	434	(65.1)	144	(21.6)	89	(13.3)
**Anxiety**	474	(71.3)	131	(19.7)	60	(9.0)
**Drowsiness**	403	(60.2)	145	(21.7)	121	(18.1)
**Appetite loss**	328	(48.9)	135	(20.1)	208	(31.0)
**Wellbeing**	379	(56.8)	168	(25.2)	120	(18.0)
**Dyspnea**	549	(83.1)	64	(9.7)	48	(7.3)

**Table 3 cancers-14-02568-t003:** Summary results of the indicators by procedure.

Appropriateness Indicators	Threshold	Results
1. Referral of patients undergoing active oncological treatment *	≥90%	91.9%
2. Referral of patients with life expectancy < 6 months	≥50%	50.1%
**Process indicators**		
1. Completed referral form	≥50%	100.0%
2. Time of visit scheduling based on the referral form score	≥80%	68.0%
3. Presence of an advance care plan	≥90%	100.0%
4. Verification of home services activation **	≥90%	93.3%
**Outcome indicators**		
1. Symptom’s evaluation (ESAS Score)	≥90%	90.0%
2. Psychological support **	≥90%	100.0%
3. Nutritional support **	≥90%	100.0%
4. Patients visiting SCOC with >2 unplanned visits to emergency room	≤10%	7.8%
5. Consistency of place of death with patient’s preference ***	≥70%	69.2%

* For all patients evaluated after form adoption; ** for patients with identified needs; *** for patients who died.

**Table 4 cancers-14-02568-t004:** Actual survival for patients referred to SCOC according to the year (2018–2021).

Time	2018		2019		2020		2021		*p*-Value
	*n*	(%)	*n*	(%)	*n*	(%)	*n*	(%)	
<6 months	79	(40.3)	66	(38.4)	87	(52.7)	145	(65.9)	<0.0001
6–12 months	50	(25.5)	51	(29.6)	35	(21.2)	67	(30.5)	0.0177
>12 months	67	(34.2)	55	(32.0)	43	(26.1)	8	(3.6)	<0.0001

**Table 5 cancers-14-02568-t005:** Scheduled visit time *.

Time Expected for the Visit	Form Score					
	≥10		5–9		0–4	
	N pts	(%)	N pts	(%)	N pts	(%)
**Within 15 days**	96	(75.0)				
**Within 1 months**	22	(17.2)	309	(64.8)		
**Within 2 months**	7	(5.5)	112	(23.5)	51	(77.3)
**Beyond**	3	(2.3)	56	(11.7)	15	(22.7)

* Out of 671 evaluable patients. Boxes are green if visit timing was consistent with referral forms’ score, and red if they were not consistent.

**Table 6 cancers-14-02568-t006:** Output of SCOC visit.

Psychological Assessment	N pts (%) 753 (100.0)
Diagnosis Awareness	675 (89.6)
*Total*	*643 (95.3)*
*Partial*	*29 (4.3)*
*Not acknowledged*	*3 (0.4)*
Prognosis Awareness	610 (81.0)
*Total*	*357 (58.5)*
*Partial*	*242 (39.7)*
*Not acknowledged*	*11 (1.8)*
Psychological problems	286 (38.0)
*Depression*	*192 (67.1)*
*Anxiety*	*52 (18.2)*
*Anxiety + Depression*	*28 (9.8)*
*Other*	*14 (4.9)*
Activation psychological support path *	99 (34.6)
Psychiatric consultation required	19 (6.6)
**Nutritional assessment (MUST)**	**652 (86.6)**
0	288 (44.2)
1	117 (17.9)
2–6	247 (37.9)
Activation nutritional support path *	247 (100.0)
Weight stabilization 30 days after the SCOC visit **	113 (63.5)
**Activation of palliative care services**	**585 (77.7)**
Hospital service	91 (15.6)
Territorial service	461 (78.8)
Postponed	33 (5.6)

* For the patients with psychological or nutritional problems, as appropriate. ** For 247 patients with MUST >1.

## Data Availability

Data presented in this study are available on request from the corresponding author.

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
