# Peer review of "Simultaneous Care in Oncology: A 7-Year Experience at ESMO Designated Centre at Veneto Institute of Oncology, Italy"

_cancers, 2022, doi:10.3390/cancers14102568_

Round 1

Reviewer 1 Report

The authors report an interesting long term experience of simultaneous care in oncology. Discussion needs to be integrated by including and dissussing the following refs already reported, concerning the real life experience of simultaneous and early palliative care, both in Canada (Zimmermann BMJ Support & Palliative Care and in Italy (Bandieri et al., Annals of Oncology 2012; Bandieri et al., BMJ Support & Palliatve Care 2019; Potenza et al., BMJ Support Palliative Care 2021).

Author Response

"The authors report an interesting long term experience of simultaneous care in oncology. Discussion needs to be integrated by including and discussing the following refs already reported, concerning the real life experience of simultaneous and early palliative care, both in Canada (Zimmermann BMJ Support & Palliative Care and in Italy (Bandieri et al., Annals of Oncology 2012; Bandieri et al., BMJ Support & Palliatve Care 2019; Potenza et al., BMJ Support Palliative Care 2021)" Answer: Thank you for this suggestion, the discussion has been implemented accordingly.

Reviewer 2 Report

Thank you for giving me the opportunity to review the manuscript “Simultaneous Care in oncology: a 7-year experience at Veneto Institute of Oncology, ESMO Designated Centre, Italy“ submitted to cancers.

The authors are to be congratulated to a scientifically valid study that includes a relevant and novel research question and addresses a scientifically relevant gap in the literature. The methodology and statistical analysis are sound and the study is relevant to the research field. The manuscript is excellently written while the contributions are presented in a clear and compelling way Furthermore, the topic is of clinical interest and highlights the importance of improving good clinical practice through new organizational models.

Nevertheless, I have some comments that should help to improve the quality of the manuscript:

Abstract

Line 44-49: please always rank numbers from the highest to the lowest percentage.

Introduction

Line 117-118: patients with a very impaired performance status might choose to stop discontinue oncologic therapy or oncologic therapy might be discontinued or stopped by the responsible physician because of high toxicity. Please specify what you mean by “oncologic care“. Please explain if the patients were offered the option to discontinue oncologic therapy.

Scores

When providing the Simultaneous care referral form (Figure 1A), the authors should briefly explain why they chose to use the Karnofsky Index (KI) instead of the ECOG und why a KI of 50-60 indicates a higher need for palliative care services, as it is well known from literature that the assessment of the KI is a highly subjective matter.

Discussion

When mentioning the ESAS, please provide a brief statement about why Patient- reported outcome measures are very relevant in oncology and palliative care.

The authors state that as for the place of death, unfortunately this did not always occur where expected. However, 70% died in the desirable location which is quite good compared to international data.

Best of luck in revising your manuscript!

Author Response

Find the answers to the comments in the attached file. Thank you. 
